# Revealing the Coral Species Diversity in Xiamen Bay: Spatial Distribution of Genus *Astrogorgia* (Cnidaria, Alcyonacea, Plexauridae) and Newly Recorded Species

**Jia-Ying Liu** [1,2,†], **Yun-Pei Wang** [1,2,†], **Jing Yang** [1,2], **Yi-Jia Shih** [1,2] and **Ta-Jen Chu** [1,2,*]

1   Fisheries College, Jimei University, Jimei District, Xiamen 361021, China
2   Fujian Provincial Key Laboratory of Marine Fishery Resources and Eco-Environment, Jimei University, Jimei District, Xiamen 361021, China
*   Correspondence: chutajen@gmail.com
†   These authors contributed equally to this work.

**Abstract:** Coral reefs provide a habitat for many marine organisms and support the safety, coastal protection, well-being, and food and economic security of hundreds of millions of people. The focus on coral species diversity cannot be overemphasized. One of them, *Astrogorgia,* contains many marine natural active substances, and has important scientific research value and application prospects. Most of the current research on the active substances of the genus *Astrogorgia* is based on unidentified species, and in-depth taxonomic studies are urgently needed. A total of 1185 samples were collected from 2014 to 2021 in the waters of Xiamen Bay. Herein, the morphological identification, electronic microscopy, and gene fragment sequencing methods were used for the taxonomic study. There are three species of *Astrogorgia* identified, including *Astrogorgia lafoa*, *A. arborea*, and *A. dumbea*. Among them, *A. lafoa* and *A. arborea* are newly recorded species in the waters of China. *A. lafoa* is distributed in Qingyu Island, *A. arborea* is distributed in Wuyu Island, and *A. dumbea* is widely distributed in Baiha Reef, Qingyu Island, Wuyu Island, and Xiaobai Island. In this paper, the geographical distribution and the habits of 18 species of *Astrogorgia* are summarized, and the evolution of family and genus classification of *Astrogorgia* is discussed. The results enrich the geographical distribution information and coral species diversity records of *Astrogorgia* in China.

**Keywords:** *Astrogorgia*; newly recorded species; taxonomy; species diversity; Xiamen Bay





## 1. Introduction

Coral reefs are found in over 100 countries and territories, and while they cover only 0.2% of the seafloor, it is important that they support at least 25% of marine species and support the safety, coastal protection, well-being, and food and economic security of hundreds of millions of people [1]. Coral is the collective name for marine invertebrates within the sub-classes Hexacorallia and Octocorallia of the class Anthozoa in the phylum Cnidaria. It can provide a good habitat for numerous organisms, thus forming a unique coral reef ecosystem with high productivity [1]. Coral reef ecosystems are often referred to as "oases in a tropical marine desert" or "rainforests in the ocean" [2]. Coral reefs are environmental indicators of water quality because they can only tolerate narrow ranges of temperature, salinity, water clarity, and other water conditions [1]. According to statistics, the global coral reef area only accounts for 0.2% of the world's ocean area, but there are about 900,000 species of organisms. The value and services provided by coral reef ecosystems account for 2.85% of marine ecosystems [3], making a huge contribution to the world.

In terms of distribution, 91.9% of the global coral reef area is distributed in the Indo-Pacific region, including the Red Sea, the Indian Sea, the Southeast Asian Sea, and the Pacific Ocean [4]. The coral reef area in the South China Sea is about 37,935 km$^2$, accounting

for about 5% of the world's coral reef area and ranking the second in Southeast Asia [3]. Coral reefs have also been found off the coast of Fujian, Guangdong, Guangxi, and Hainan in China. In the past, the coral reef in Xuwen Island, Guangdong, was considered the northernmost modern coral reef off the coast of China [5]. Zou [5] believes that Dongshan in Fujian is the northernmost limit of the distribution of shallow water reef building coral communities in China.

In China, coral reefs include 80 genera and more than 700 subgenera, belonging to the Indo-Pacific biogeographic region, extending from the Taiwan Strait to the South China Sea, and are characterized by a high diversity of coral species. The Checklist of Marine Biota in China Seas edited by Liu [6] has included a total of 1422 species of cnidarians, of which 763 are species of coral resources, including 41 species of Ceriantipatharia of Anthozoa, 328 species of Octocorollia, and 394 species of *Scleractinia* of Hexacorallia. According to Huang et al. [7], reef-building corals belong to 2 groups, 16 families, 77 genera, and 445 species in China. Yang [8] analyzed that there are 54 species of gorgonians in 9 families and 22 genera in Fujian waters. Huang [9] mentioned that there are 28 species of corals in 10 families in Xiamen Bay. In 2018, Ni [10] found 16 newly recorded species of coral in Baiha Reef, Xiamen Bay, of which *Astrogorgia* was a newly recorded genus, but the species level could not be identified. In short, Xiamen Bay is a suitable sea area for corals to inhabit. There may be more coral species that have yet to be discovered due to less historical research.

*Astrogorgia* was established in 1868 [11], the type species is *A. sinensis* Verrill, 1865 [12]. The main characteristics of the genus are fan-shaped coral colonies, with irregular lateral branches, and the branches are not fused into a network [11]. The sclerites are spindle-shaped, longer than the diameter of the branches, and are red, orange, yellow, or colorless [11]. Corals are dark red, red, pink, orange, yellow, cream, or tan [11,13]. At present, there are 18 valid species of this genus on WROMS (https://www.marinespecies.org/, accessed on 2 June 2022) [14]. Species of this genus are widely distributed in Indonesia, New Caledonia, and Japan [15]. Two species of this genus, *A. dumbea* [16] and *A. sinensis* [12,17], have been reported in the offshore waters of China. Among them, *A. dumbea* inhabits Dongshan Bay, Fujian, and *A. sinensis* inhabits the offshore waters of Hong Kong and Taiwan.

Some studies on the genus *Astrogorgia* mainly focus on the following two aspects: one is the natural active substances, and the other is the taxonomic identification and geographical distribution. The first part includes the extraction of substances [18], the structure and biological activity of substances [15,18], the physiological and biochemical functions, and effects of natural active substances [19–22]. Kandou et al. [23] found that the active substances of *Astrogorgia* sp. have bactericidal or bacteriostatic properties. Some *Astrogorgia* also contains alkaloids, steroids, and other chemicals with antibacterial, antioxidant, and anti-cytotoxic effects [24]. Diterpenoid secondary metabolites extracted from *Astrogorgia* can also defend against natural enemies and reduce the chance of predation [25–28]. Most of the above studies on the active substances are based on the undetermined species of this genus, which is not conducive to the research and production application in the future. Therefore, it is crucial to conduct in-depth taxonomic research on the species of this genus.

The other part is the identification of *Astrogorgia* [29]. It is mainly based on the morphological characteristics of *Astrogorgia* [30]. The phenomenon of synonymy is due to a number of reasons [30–32], including phenotypic plasticity [33], detailed identification and a lack of geographic distribution information [31], and an inconsistent basis for species identification [34–36]. In recent years, there have been great changes in the taxonomic classification of *Astrogorgia* species [31,32]. At the same time, with the rapid development of molecular biology technology, gene fragment sequencing is also widely used in the classification and identification. Some scholars show that the COI, ND2, and MSH gene fragments are suitable for the phylogenetic study of *Octocorallia* [37–39].

It is very important to classify and identify corals on the basis of morphology combined with molecular biology techniques. Xu [40] established a phylogenetic tree of mtMutS

genes based on morphology and identified coral species of the family Chrysogorgiidae and the diversities of seamounts in the tropical Western Pacific. When discussing the molecular phylogenetic relationship of three species of gorgonians in the South China Sea, Li et al. [41] established the NJ and MP phylogenetic tree, and the phylogenetic tree clearly shows the phylogenetic relationship of corals.

This study mainly used external morphology and gene fragment technology to classify and identify coral samples collected from the waters in Xiamen Bay. The aim of this research is to explore the geographic distribution and coral species diversity of *Astrogorgia.* The result can provide references for in-depth classification and suggestions for formulating management policies and the conservation of coral resources.

## 2. Materials and Methods

### 2.1. Sample Collection and Study Area

From 2014 to 2021, a total of 1185 samples were collected by diving in Xiamen Bay. Among them, 57 samples were collected from Baiha Reef in 2014, 96 samples were collected from Baiha Reef and Jiaoyu Island in 2015, 104 samples were collected from Baiha Reef and Jiaoyu Island in 2016, 823 samples were collected from Baiha Reef, Jiaoyu Island, Shangyu Island, Huangcuo, Kulangsu Island, Fire Island, Dabai Island, Xiaobai Island, Qingyu Island, and Wuyu Island in 2017, and 105 samples were collected from Dabai Island, Xiaobai Island, Qingyu Island, and Wuyu Island in 2021. There were 17 samples of genus *Astrogorgia.* The sampling stations' information is shown in Table 1, and the map of the sampling stations is shown in Figure 1.

**Table 1.** Sampling stations' information.

| Year | Station | Longitude | Latitude |
|------|---------|-----------|----------|
| 2014 | Baiha Reef (BH) | 118°22′06″–118°22′17″ | 24°31′39″–24°31′55″ |
| 2015 | Jiaoyu Island (JY) | 118°24′15″–118°24′40″ | 24°32′42″–24°33′22″ |
| | Baiha Reef (BH) | 118°22′12″–118°22′17″ | 24°31′53″–24°31′55″ |
| 2016 | Jiaoyu Island (JY) | 118°24′15″ | 24°33′40″ |
| | Baiha Reef (BH) | 118°22′10″–118°22′17′ | 24°31′45″–24°31′51″ |
| | Shangyu Island (SY) | 118°11′19″–118°11′25″ | 24°27′12″–24°27′13″ |
| | Huangcuo (HC) | 118°07′50″ | 24°25′25″ |
| | Kulangsu Island (KLS) | 118°03′14″–118°03′41″ | 24°26′23″–24°26′35″ |
| | Fire Island (FI) | 118°03′54″ | 24°29′35″ |
| 2017 | Qingyu Island (QY) | 118°05′35″–118° 07′28″ | 24° 21′45″–24°21′55″ |
| | Wuyu Island (WY) | 118°03′54″–118° 08′35″ | 24° 20′23″–24° 29′35″ |
| | Dabai Island (DB) | 118°26′58″–118°27′40″ | 24°33′46″–24°34′9″ |
| | Xiaobai Island (XB) | 118°27′47″ | 24°33′21″ |
| | Jiaoyu Island (JY) | 118°24′14″ | 24°32′41″ |
| | Baiha Reef (BH) | 118°22′07″–118°22′17″ | 24°31′38″–24°31′55″ |
| 2021 | Qingyu Island (QY) | 118°07′22″–118°07′50″ | 24°21′46″–24°22′11″ |
| | Wuyu Island (WY) | 118°08′29″–118°08′57″ | 24°20′30″–24°20′51″ |
| | Dabai Island (DB) | 118°45′01″–118°46′06″ | 24°56′32″–24°57′51″ |
| | Xiaobai Island (XB) | 118°42′59″–118°43′39″ | 24°58′26″–24°58′96″ |

Live coral samples were collected and brought back to the laboratory, where they were numbered and photographed. The samples were stored in 95% ethanol solution or a −20 °C refrigerator. Subsequently, the samples were used for microscopic morphology observation and molecular biological analysis. The processed samples were deposited in the Coral Ecology Laboratory of Fisheries College, Jimei University.

### 2.2. Sample Processing and Observation

First, the appearance, color, coral polyps, surface, main axis and branching, the base and the top of the colony characteristics, etc., were observed. The relevant parts of the coral were measured, photographed, described, and recorded. The color and shape of polyps and axial bones were observed under a dissecting microscope (Nikon SMZ1270, Fujian

Provincial Key Laboratory of Marine Fishery Resources and Eco-Environment) and photos were taken.

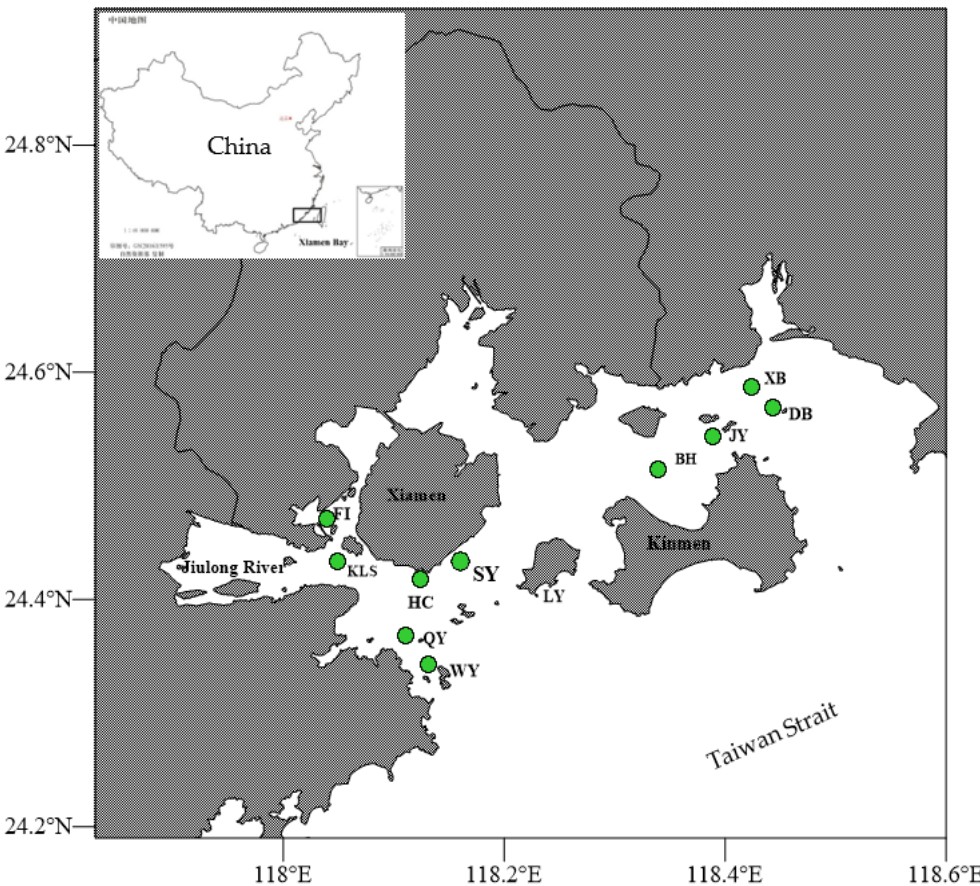

**Figure 1.** Map showing the location of the sample stations (FI, KLS, HC, SY, QY, WY, BH, JY, XB, and DB) in Xiamen Bay.

Sclerite samples from different parts of the colonies (polyps, coenenchyme, tentacles) were examined separately after dissolving tissue samples in sodium hypochlorite. The sclerites were washed with distilled water and dried with a blast dryer, mounted for the scanning electron microscope (SEM) (G5 PhenomProX2017) from National Key Laboratory of Xiamen University. Detailed images of the sclerites were obtained with SEM using Adobe Photoshop 2020 and optimum magnification for each sclerite type.

Coral branches were placed in a petri dish and dissolved in a ratio of 3:1–5:1 of ultrapure water and sodium hypochlorite solution. The arrangement of the spicules was observed and photographed under a dissecting microscope.

### 2.3. Species Identification

#### 2.3.1. Morphology

The identification method refers to the original literature of species [42,43], "Soft corals and Sea Fans" [13], the World Register of Marine Species (WROMS) [14], and the taxonomic term reference [44].

#### 2.3.2. DNA Extraction and PCR Amplification

Three individual corals were prepared for total genomic DNA extraction using the marine animal tissue genomic DNA extraction kit, Shanghai Biotechnology Co., Ltd. The steps of the extraction followed the manufacturer's instructions.

The COI gene was selected for genetic analysis. The primers used in this analysis are presented as follows: COI-LA-8398: F-GGA ATG GCG GGG ACA GCT TCG AGT ATG

TTA ATA CGG, and COIoct: R-ATC ATA GCA TAG ACC ATACC [45]. Each reaction was performed in a 25 uL solution, containing: upstream primer 2.5 uL, downstream primer 2.5 uL, template 2 uL, and water 18 uL. PCR reaction conditions were: denaturation at 94 °C for 3 min, followed by 35 cycles of denaturation at 94 °C for 30 s, annealing at 55 °C for 30 s, extend at 72 °C for 1 min, and end the cycle with a final extension of 7 min at 72 °C. PCR amplification products were electrophoresed on a 1% agarose gel, stained with EB (Ethidium Bromide), and visualized on a gel meter. After the PCR products were electrophoresed on a 1% gel, the positive products were selected and sent to Shanghai Biotechnology Co., Ltd. for sequencing.

### 2.3.3. Cladogram Topology

Three fragments of the COI gene were obtained from individual *A. lafoa*, *A. arborea*, and *A. dumbea* and those sequences were deposited in GenBank. The reference sequences of interspecies and outgroups were selected for phylogenetic analysis from the GenBank database. The list of species and the GenBank accession numbers of their DNA fragments are presented in Table 2. The sequence of the COI gene was separated for analysis and topology, respectively. First, the sequences were aligned using the CLUSTAL-W program by Bio Edit version 7.2.3. Subsequently, neighbor-joining (NJ) analyses of the dataset were performed using the MEGA7.0 software and produced cladogram topology. The neighbor-joining method was based on the Kimura two-parameter (K2P) model with 1000 bootstrap replicates.

**Table 2.** GenBank accession numbers and reference species in this study.

| Number | Species | GenBank Accession Number COI |
|:---:|:---:|:---:|
| 1 | *Astrogorgia* sp. | KF955026.1 |
| 2 | *Astrogorgia* sp. | KF955025.1 |
| 3 | *Astrogorgia* sp. | KF955022.1 |
| 4 | *Astrogorgia* sp. | KF955021.1 |
| 5 | *Astrogorgia* sp. | KF955020.1 |
| 6 | *Astrogorgia fruticosa* | MT724668.1 |
| 7 | *Astrogorgia* sp. | KF955029.1 |
| 8 | *Astrogorgia* sp. | KF955028.1 |
| 9 | *Astrogorgia* sp. | KF955024.1 |
| 10 | *Astrogorgia* sp. | KF955023.1 |
| 11 | *Astrogorgia* sp. | KF955027.1 |
| 12 | *Astrogorgia* sp. | KF955030.1 |
| 13 | *Astrogorgia* sp. | JX203861.1 |
| 14 | *Astrogorgia* sp. | GQ342444.1 |
| 15 | *Astrogorgia lafoa* * | ON748930 |
| 16 | *Astrogorgia dumbea* * | ON748931 |
| 17 | *Astrogorgia arborea* * | ON748932 |
| 18 | *Muricea laxa* | MK153426.1 |

Note: * indicates that the sequences were obtained in this study.

## 3. Results

### 3.1. Systematic

#### 3.1.1. *Astrogorgia lafoa* Grasshoff, 1999

*A. lafoa* Grasshoff, 1999: (Grasshoff; 1999: 43; Alderslade and Fabricius, 2008: 210) [13,42].
Material examined: QY (118°07′34″, 24°21′46″), 7.5 m, 23 April 2021.
Research sample number: 20210423-QY-02-14.
Description: The living colonies are flat fan-shaped (Figure 2A), up to 40 mm high, and the base is 10 mm long. The stem and branches are cylindrical. The stem is thicker, up to 3 mm in diameter. The new branches form an acute angle with the original branch, and the branches are discrete but not fused and connected. Branches grow outward and extend in diameter, gradually decreasing, with a terminal diameter of 0.5 mm.

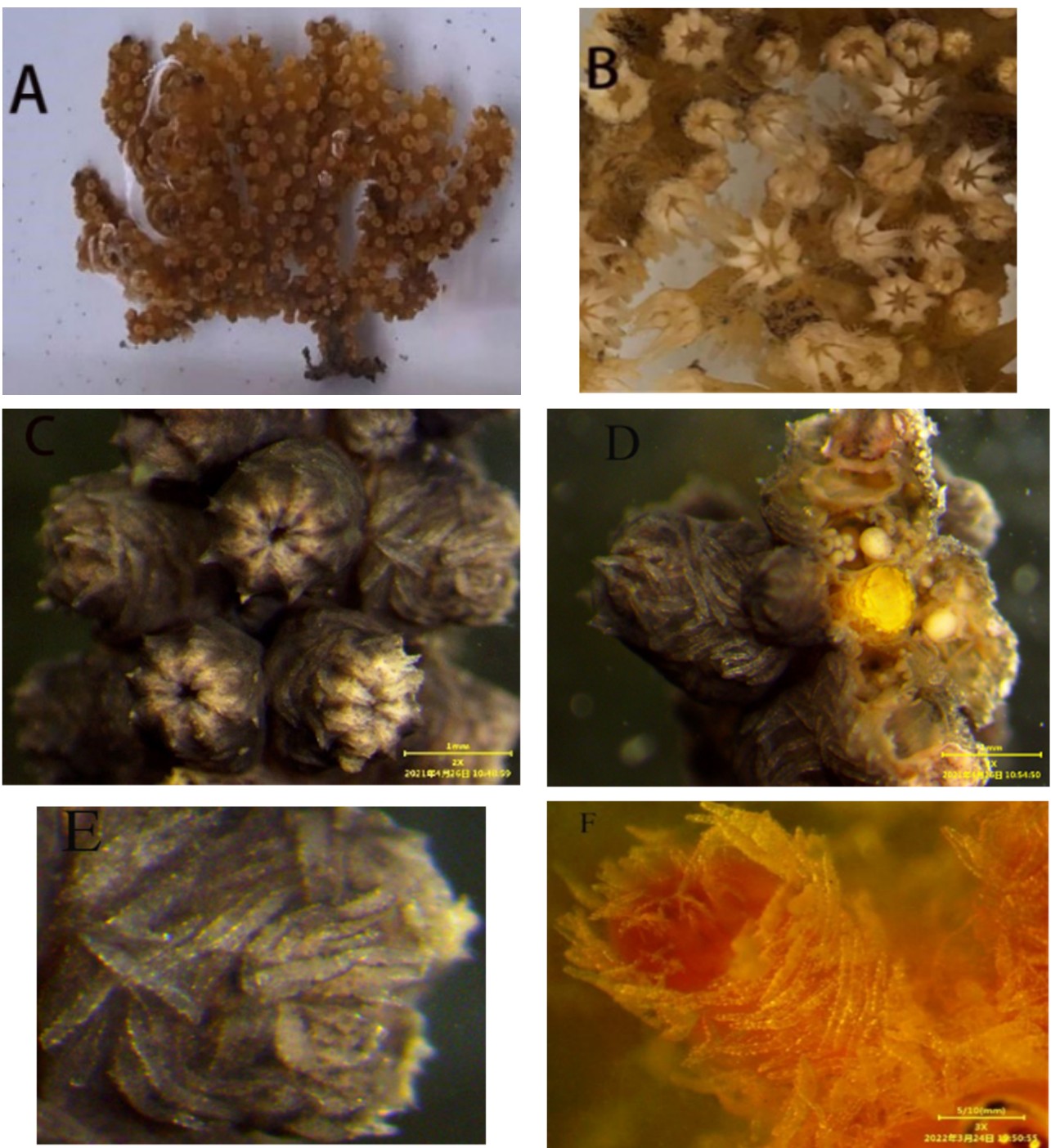

**Figure 2.** Map showing the external morphology of *A. lafoa*, axis, and polyps. (**A**) Live colony, (**B**) a detail of a colony, (**C**) polyps, (**D**) axis, and (**E,F**) polyps sclerites.

Coral group: The coral body is flat fan-shaped, the new branches and the original branches are at an acute angle, and the branches are not fused and connected. The coral body is 4 cm high, and the base is 1 cm long. The stem is thicker, the diameter can reach 0.3 cm, and the diameters of other branches are in order. It gradually becomes smaller, and the diameter of the end is 0.05 cm. The stem and branches are cylindrical.

Polyps: Polyps monotype, distributed on the stem and branch surfaces. When the polyps contract, they are cone-shaped protrusions with an interval of about 1 mm. The

polyps are about 1 mm in diameter, the coral calyx is obvious, and the coral calyx is about 1 mm high.

Tentacle: Anthocodia, with large bent sclerites 0.25 mm long and rods 0.15–0.20 mm long at the bases of the tentacles.

Sclerite: Both the polyp tentacles and the polyps themselves contain spindle-shaped sclerites (Figure 3). The sclerites of the tentacles are about 0.2–0.5 mm long and have sparse conical protrusions on the surface, while the polyp sclerites are about 0.2–0.8 mm long and have many warts on the surface. The sclerites in the coenenchyma are mostly spindle-shaped and partially deformed. The sclerites are large in size, about 0.2–1 mm in length, and up to 1 mm in width. There are many large wart-like protrusions on the surfaces of the sclerites.

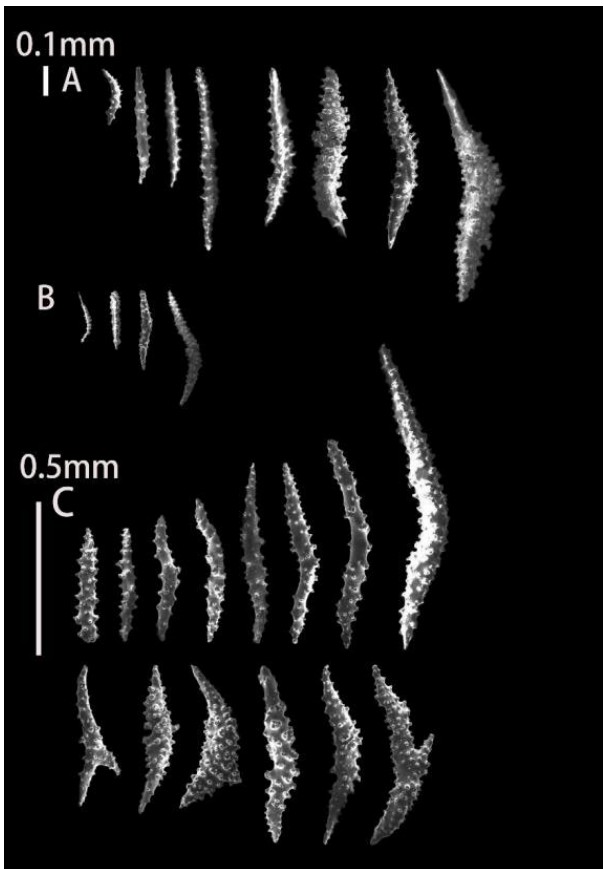

**Figure 3.** Map showing the sclerites of *A. lafoa*, axis, and polyps. (**A**) Sclerites of polyp, (**B**) sclerites of polyp tentacle, and (**C**) sclerites of coenenchyma. Scales: A = B = 0.1 mm. C = 0.5 mm.

Colony color: bright orange, dark brown in alcohol.

Sample collection site: the waters near Qingyu Island in Xiamen Bay, with a water depth of 7.5 m (118°07′34″, 24°21′46″).

3.1.2. *Astrogorgia arborea* (Thomson & Simpson, 1909)

*A. arborea* Thomson & Simpson, 1909: (Thomson & Simpson, 1909: 255–257; Alderslade and Fabricius, 2008: 210) [13,43].

Material examined: WY (118°8′57″, 24°20′51″), 5 m, 23 April 2021.

Research sample number: 20210423-WY-03-18.

Description: The living colonies are scattered branches (Figure 4A), up to 60 mm high, and the base is 14 mm long. The stem and branches are cylindrical. Four branches were observed on the same base. Three branches were bi-branched, and the other one was not

yet branched. The ends of the branches are swollen. The stems are about 3 mm in diameter and the ends are 7 mm.

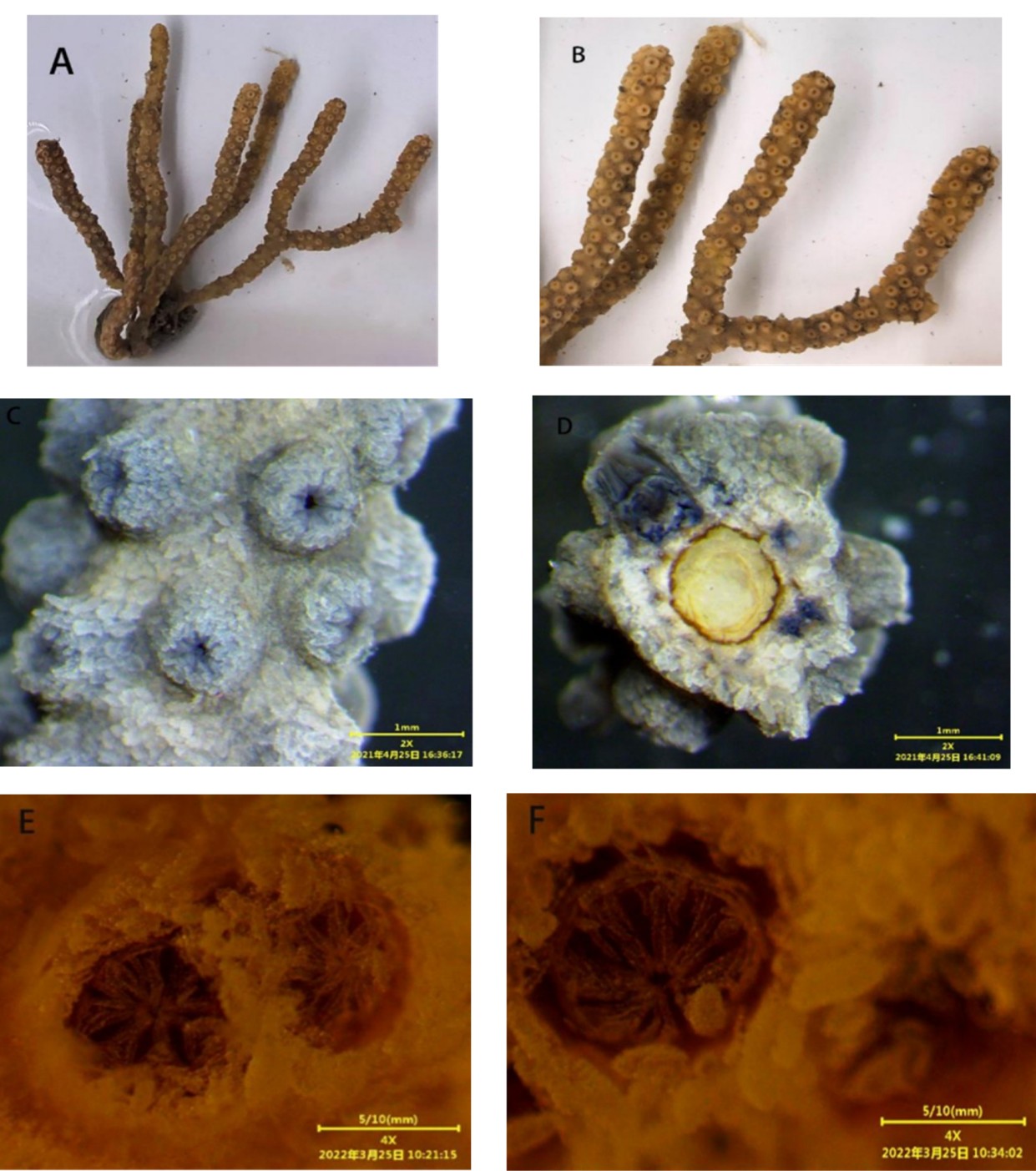

**Figure 4.** Map showing the external morphology of *A. arborea*, axis, and polyps. (**A**) Live colony, (**B**) a detail of a colony, (**C**) polyps, (**D**) axis, and (**E**,**F**) polyps sclerites.

Polyps: Polyps monotype, distributed on the stem and branch surfaces. Polyps can completely shrink into coenenchyma. The polyps are about 1 mm in diameter. The coral calyx is about 0.5 mm high and has a circular diameter of about 1 mm.

Tentacle: The tentacles are densely covered on the aboral surface with small spindle shapes. These are arranged "en chevron" at the base, but longitudinally higher up. When

the tentacles are unfolded, they touch on the aboral surfaces, and no definite operculum is formed.

Sclerite: Spindle-shaped sclerites are densely distributed on the surface of polyps and coenenchyma tissue (Figure 5). Sclerites are slender, about 0.18–0.2 mm. There are many small wart-like protrusions on the surface. The sclerites of the coenenchyma are mainly spindle-shaped, and some are abnormal. The sclerites are relatively thick, about 0.1–0.2 mm long. The surface is densely covered with large wart-like protrusions. Coral calyx sclerites are arranged in an "en chevron" shape with gradually decreasing angles.

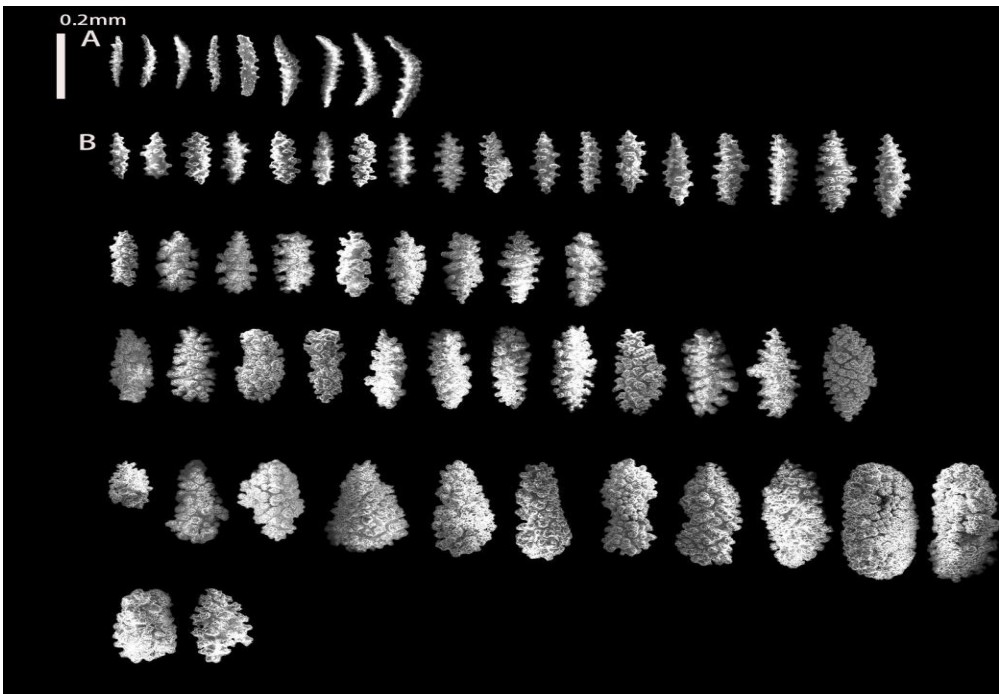

**Figure 5.** Map showing the sclerites of *A. arborea*, axis, and polyps. (**A**) Sclerites of polyp, (**B**) sclerites of coenenchyma. Scales: A = B = 0.2 mm.

Colony color: brownish yellow, dark brown in alcohol.

Sample collection site: the waters near Wuyu Island in Xiamen Bay, with a water depth of 5 m (118°8′57″, 24°20′51″).

### 3.1.3. *Astrogorgia dumbea* Grasshoff, 1999

*A. dumbea* Grasshoff, 1999: (Lee et al., 2011; Grasshoff, 1999: 38; Alderslade and Fabricius, 2008: 210) [13,21,42].

Material examined: QY (118°07′34″, 24°21′46″), 7.5 m, 23 April 2021.

2014-BH-01, 2014-BH-02, 2015-BH-01, 2017-QY-01-01, 2017-QY-01-02, 2017-QY-01-03, 2017-QY-02-01, 2017-WY-01-01, 2017-WY-01-02, 2017-WY-01-03, 2017-WY-01-04, 2017-WY-02-01, 2017-WY-02-02, 2021-XB-01-01, 2021-XB-01-02, 2021-XB-01-03, 20210423-QY-02-15.

Research sample number: 20210423-QY-02-15.

Description: Coral colonies are fan-shaped (Figure 6A), up to 11 cm high, and the base is 11 mm long. The new branches are at right angles to the original branches and extend upward. The branches are not exactly in the same plane. The stem is thick, and the branches are sparse and short. The diameter of the stem and the branches are basically the same, about 0.2 cm, and the branches are laterally flat.

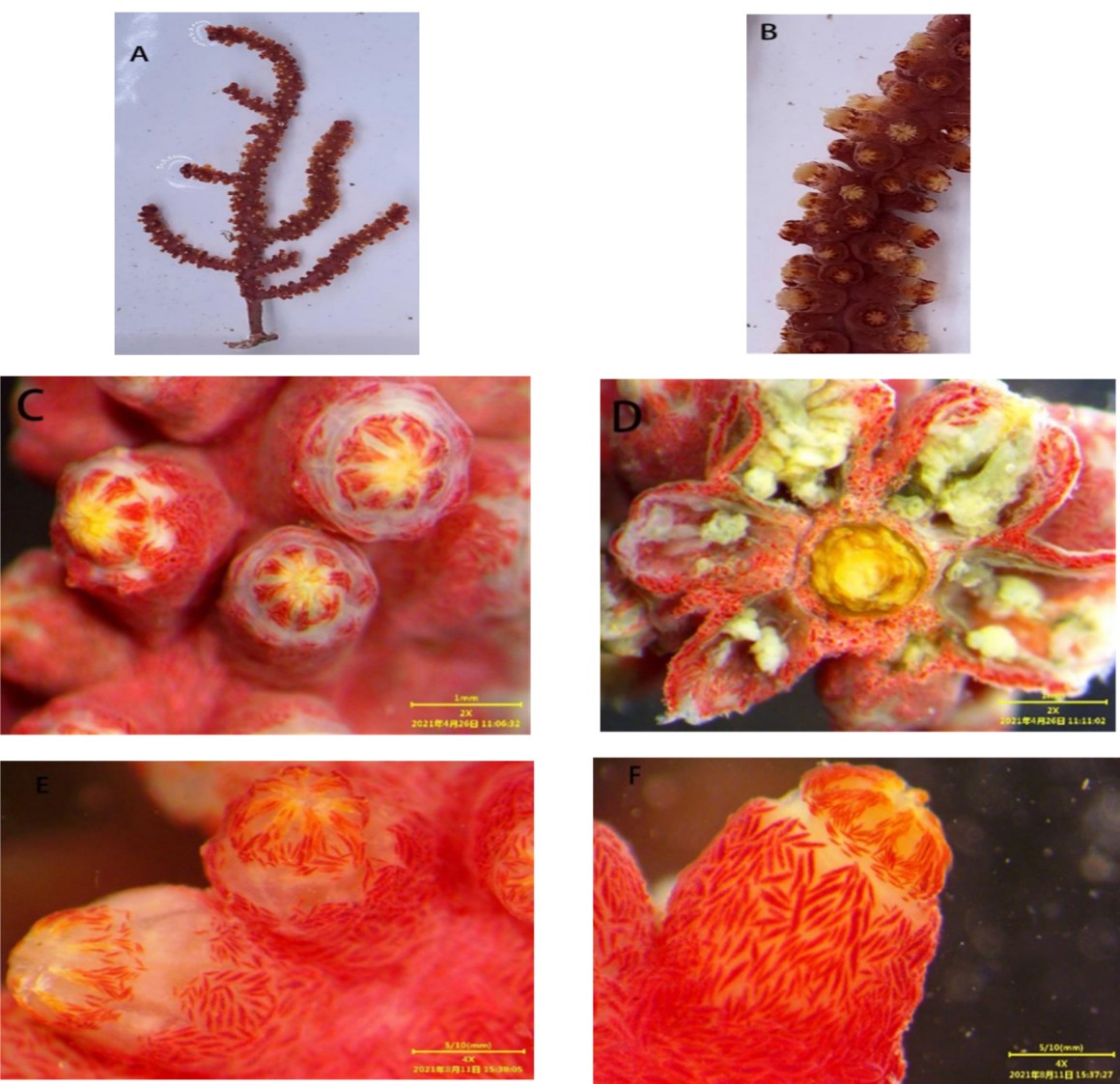

**Figure 6.** Map showing the external morphology of *A. dumbea*, axis, and polyps. (**A**) Live colony, (**B**) a detail of a colony, (**C**) polyps, (**D**) axis, and (**E,F**) polyps sclerites.

Polyps: Polyp monotype, distributed around the stem and branches, but with a tendency to concentrate on the sides. The polyps are about 1 mm in diameter. The polyps shrink slightly higher than the coral calyx. The coral calyx is obvious and is about 1 mm high.

Tentacle: Tentacles with numerous small sclerites in oblique double rows. Color is bright red with white tentacles.

Sclerite: Polyps and tentacles have spindle-shaped sclerites (Figure 7). The tentacle sclerites are about 0.1–0.3 mm long. The polyps sclerites are about 0.2–0.3 mm long and have warty protrusions on the surface, with an "en chevron" shape. The sclerites of the coenenchyma are mostly spindle-shaped. The smaller sclerites are about 0.12–0.15 mm long, and the larger ones are about 0.3–0.5 mm. The surface of the larger sclerites has warty protrusions of different sizes.

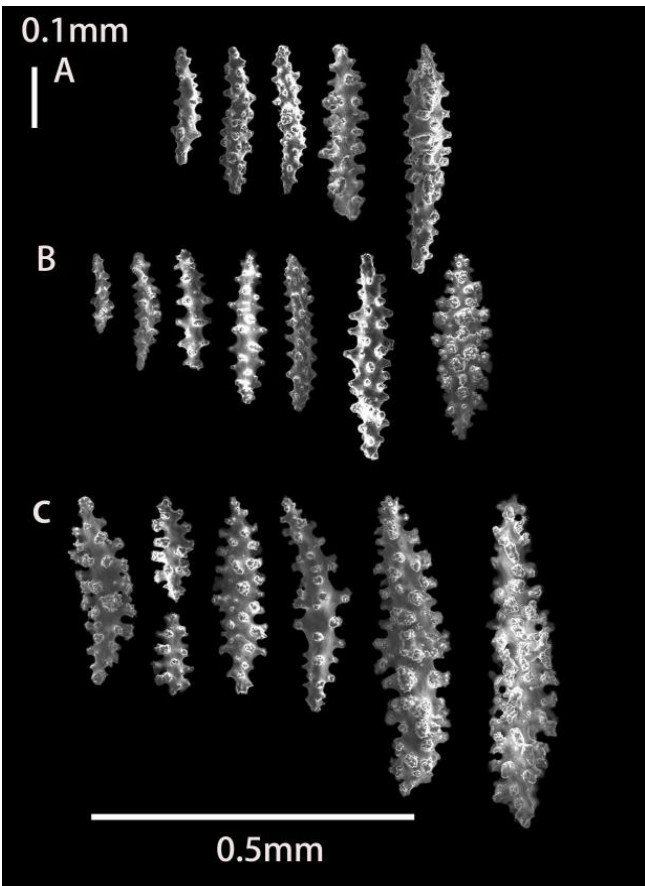

**Figure 7.** Map showing the sclerites of *A. dumbea,* axis, and polyps. (**A**) Sclerites of polyp, (**B**) sclerites of polyp tentacle, and (**C**) sclerites of coenenchyma. Scales: A = B = 0.1 mm, C = 0.5 mm.

Colony color: red, also red in alcohol and brick red when dry.

Sample collection site: the waters near Qingyu Island in Xiamen Bay, with a water depth of 7.5 m (118°07′34″, 24°21′46″).

### 3.2. DNA Analysis

A cladogram of the COI gene was established as shown in Figure 8. According to the cladograms, the *Astrogorgia* spp., *A. fruticose,* and *A. dumbea* (20210423-QY-02-15) were shown to share one clade, but the specimens *A. lafoa* (20210423-QY-02-14) and *A. arborea* (20210423-WY-03-18) were significantly separated. All of them converged to a main clade and were independent to the outgroup *Muricea laxa*. However, specimens *A. lafoa* and *A. arborea* seem to be far apart in terms of *A. fruticose,* and the results may be limited by the lack of reference sequences of congener species in this study. Hence, we still considered that the three specimens belonged to the genus *Astrogorgia* in this study.

### 3.3. Geographical Distribution and Habit Characteristics of Astrogorgia

The number of genera of corals inhabiting at each station is shown in Figure 9. The BH, DB, WY, and XB sites are the areas with relatively high coral species diversity in Xiamen Bay. The four stations are relatively far from areas of human activities.

Species of genus *Astrogorgia* are widely distributed in Indonesia, New Caledonia, and Japan [15]. By referring to the literature of *Astrogorgia* and related information on WROMS, the species list, distribution, and habit characteristics of *Astrogorgia* are summarized as shown in Table 3.

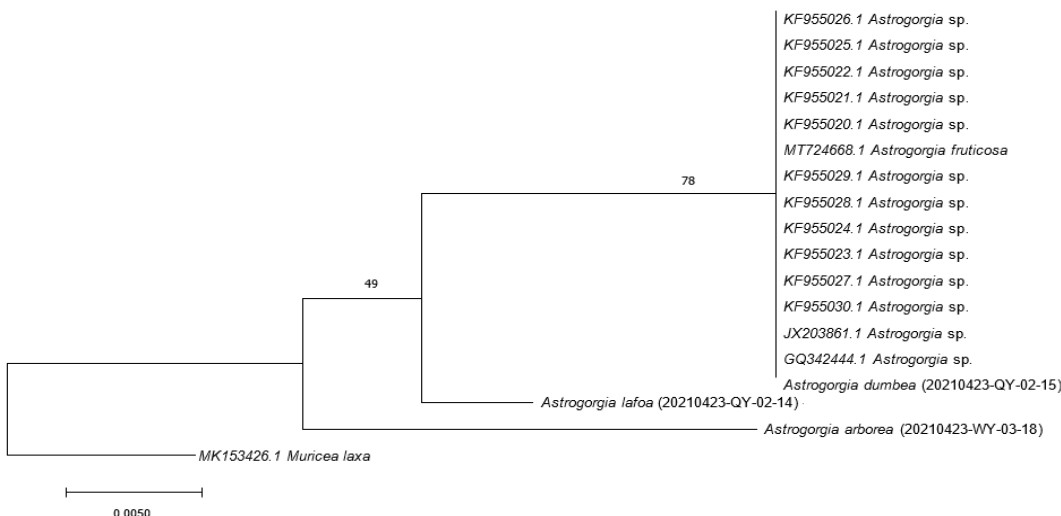

**Figure 8.** The cladogram topology of COI fragments.

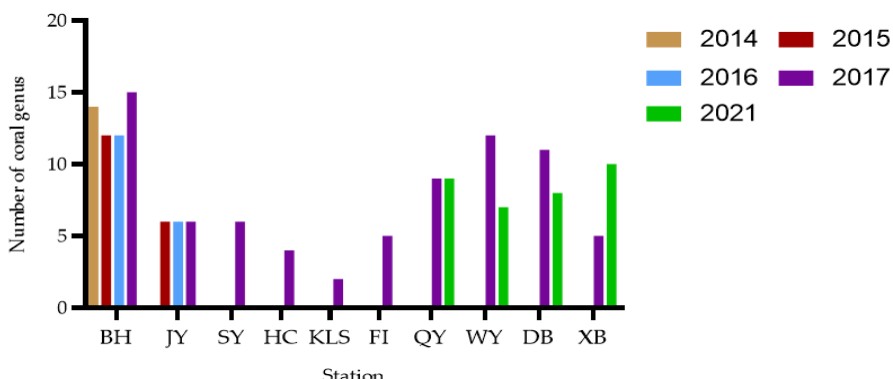

**Figure 9.** Number of coral genera at each station from 2014 to 2021.

**Table 3.** Species list and distribution information of *Astrogorgia*.

| Number | Species | Distribution Area (Reference) |
|---|---|---|
| 1 | *A. arborea* | Indian Ocean [43] |
| 2 | *A. balinensis* | Bali [46] |
| 3 | *A. bayeri* | South Sulawesi [47] |
| 4 | *A. begata* | New Caledonia, the lagoon at 25–35 m [42] |
| 5 | *A. canala* | New Caledonia, 10–30 m [42] |
| 6 | *A. dumbea* | New Caledonia Lagoon 15–45 m [42]; Dongshan Bay, China [16] |
| 7 | *A. filigella* | Japan [48] |
| 8 | *A. fruticosa* | Some shallow waters in the Persian Gulf [49] |
| 9 | *A. jiska* | Sinai Coast and the Strait of Gubar in the Red Sea [50] |
| 10 | *A. lafoa* | New Caledonia, Cape St. Vincent near 70 m [42] |
| 11 | *A. lea* | Sinai coast and the Red Sea Gulbal Strait [50] |
| 12 | *A. mengalia* | New Caledonia, Eastern Reef 35–45 m [42] |
| 13 | *A. milka* | Sinai coast and the Red Sea Gulbal Strait [50] |
| 14 | *A. ramosa* | Andaman Sea, Indian Ocean 83–494 m deep [43]; |
| 15 | *A. rubra* | Deep sea [51] |
| 16 | *A. sara* | Sinai Coast and the Strait of Gubar in the Red Sea [50] |
| 17 | *A. sinensis* | Hong Kong, China [12]; Taiwan, China [17] |
| 18 | *A. splendens* | Indian Ocean [43] |

There two species of *Astrogorgia*, *A. dumbea* [16] and *A. sinensis* [12,17], that have been reported in the offshore waters of China. Among them, *A. dumbea* inhabits Dongshan Bay, Fujian, and *A. sinensis* inhabits the offshore waters of Hong Kong and Taiwan.

Previously, there was no report on the habitat of genus *Astrogorgia* in Xiamen Bay. There are three species that are newly recorded species of *Astrogorgia* in Xiamen Bay, including *A. lafoa*, *A. arborea,* and *A. dumbea*. More importantly, *A. lafoa* and *A. arborea* are newly recorded species in the neritic area of China.

There are two species of *Astrogorgia* distributed in Qingyu Island, *A. lafoa* and *A. dumbea,* and there are two species of *Astrogorgia* distributed in Wuyu Island, *A. arborea* and *A. dumbea.* In Baiha Reef and Xiaobai Island station, only one species, *A. dumbea,* is distributed, while no *Astrogorgia* species is distributed in other stations (Figure 10).

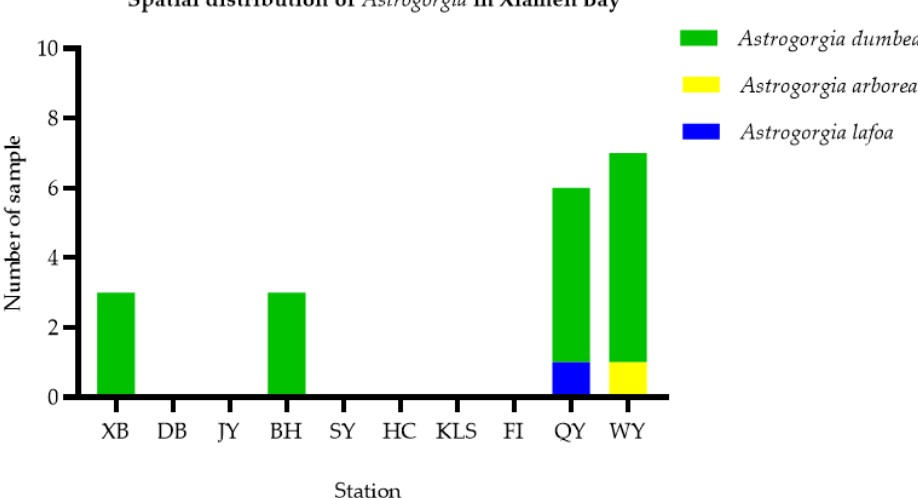

**Figure 10.** Spatial distribution of *Astrogorgia* in Xiamen Bay.

## 4. Discussion

### 4.1. Historical Evolution Process of Family-Level and Genus-Level Taxonomy of Some Species of Astrogorgia

The family-level classification of some species of the genus *Astrogorgia* is clearly under Plexauridae after long-term evolution observation. Family Plexauridae was established by Gray in 1859 [52], and Acanthogorgiidae was also established by Gray in 1859 [53]. The morphological characteristics of these two families are similar. The species of *A. arborea* in Plexauridae was originally classified into the genus *Muricella* of the family Acanthengorgonian. Later, according to the morphological comparison, *M. arborea* of the genus *Muricella* of the family Acanthengorgonian was confirmed to be *A. arborea* of the genus *Astrogorgia* [43].

Bayer established Paramuriceidae in 1956 [54]. However, there is a phenomenon of crossover in the characteristics of the Family Plexauridae and the Family Paramuriceidae. The *A. ramosa* was originally classified into the genus *Elasmogorgia* of Family Paramuriceidae by Kinoshita and Aurivillius [54] according to the morphological characteristics. Later, *E. ramosa* of the genus *Elasmogorgia* was identified as *A. ramosa* of the genus *Astrogorgia* [43] based on the morphological comparison by Aurivillius [32].

There is also confusion in the taxonomy between *Astrogorgia* and other genera of the same family, involving *Eumuricea* and *Muricea*. *Eumuricea* was first established by Verrill in 1869 [11], *Astrogorgia* was first established by Verrill in 1868 [11], and *Muricea* was first established by Lamouroux in 1821 [55]. There are similarities in the morphological characteristics of these three genera. In some studies, the essential characteristic of the genus *Muricea* is that this genus has elongated cylindrical coral calyxes with truncated edges. The genus *Eumuricea* is not sufficiently different from the genus *Muricea* to be considered a separate genus. The only useful feature to distinguish the two genera is

*Eumuricea's* tubular coral calyx, which cannot be a distinguishing feature between them [56]. Among the *Astrogorgia*, *A. splendens* and *A. ramosa* were originally classified as *Eumuricea* by Deichmann. The species *E. splendens* and *E. ramosa* in the genus *Eumuricea* were later renamed *A. splendens* and *A. ramosa* [14].

*4.2. Species Identification of Astrogorgia from Two Aspects of Morphology and Molecular Biology*

At present, the identification of coral species has traditionally been mainly based on external morphological characteristics of the coral, such as external characteristics of the colony (color, branching, size, and distribution of coral calyx), polyp (form when expanding and contracting), and spicules characteristics (shape, size, and color), which were compared and identified in combination with the original literature. However, this is often difficult to distinguish among coral species with many homologous traits and hybrids. Due to the lack of clear classification and identification standards for coral species, and the characteristics of corals such as plasticity and homogeneity, the study of coral diversity is limited, and the identification of coral species has certain challenges. Quattrini et al. [57] utilized two widely used DNA barcode markers, mtMuts and 28SrDNA, for species identification. The identification results were inconsistent with the morphological results. The established phylogenetic tree showed that the three morphologically distinct species could not be distinguished by DNA barcoding. Studies have demonstrated that DNA barcoding of mitochondrial genes is useful in the identification of many populations [58], but none of the single-gene molecular barcodes currently used to identify octopus can fully identify all species [59,60]. Huang et al. [61] used the mitochondrial COI gene sequence and a non-coding region sequence to study the phylogenetic relationship of 41 species of Coralidae in the sea near Singapore. Arrigoni et al. [62] used COI as a molecular marker to conduct a phylogenetic analysis of coral species in the family Coralidae in the Indian Ocean. The analyses by these scholars found that there are certain differences in the molecular phylogenetic results and morphological results of the hive coral family, which proves that stony corals commonly exist in the phenomenon of parallelism. Li et al. [63] conducted a comparative analysis of the COI gene fragment sequences of eight species of stony corals in Xuwen area through specific amplification and sequencing of the COI gene. The results show that molecular phylogenetic classification is slightly different from traditional morphological classification results. Coral phenotypic plasticity may have an impact on traditional classification. Therefore, the classification and identification of corals needs to combine morphology and molecular biology.

With the development of molecular biotechnology, the classification of coral species has become more and more common. For the identification of genera and species using coral gene fragment sequencing, some conditions must be considered and prepared, including aligning with the corresponding gene sequences with more molecular data in the NCBI database, and then combining the coral sequences in GenBank to construct a phylogeny tree.

As of 30 April 2022, there are 56 pieces of molecular sequence data in the NCBI database, including 1 COI gene sequence and 4 MSH gene sequences of *A. fruticosa*, and the rest are the gene fragment information of undetermined species in *Astrogorgia*. There are few molecular data in the NCBI database, so to identify the other 17 species, except *A. fruticosa*, it is necessary to analyze the detailed internal and external morphological and structural characteristics of the samples. In this study, by constructing a phylogenetic tree of COI gene fragments, the coral samples were determined to be of the genus *Astrogorgia*. Species were then identified by morphological characteristics of coral samples. To enrich the molecular database of *Astrogorgia*, the different gene sequences need further study.

*4.3. Distribution and Habitat Analysis*

According to the summary of the species list and distribution, it was shown that the *Astrogorgia* has a wide distribution range, including species inhabiting the deep seas of the Indian Ocean and the Red Sea, as well as species inhabiting the shallow sea in New Caledonia, the Persian Gulf, and Dongshan Bay and Xiamen Bay in China. Most *Astrogorgia*

species are distributed in shallow waters, such as *A. lafoa*, *A. arborea,* and *A. dumbea*, etc. Few species such as *A. ramosa* are distributed in the deep-sea waters of 83–494 m in the Indian Ocean, and *A. rubra* also inhabits the deep-sea waters.

Water temperature is an important factor affecting the biological physiology of coral. Generally speaking, the most suitable water temperature for the growth of reef-building coral is 23–28 °C, and the water temperature below 18 °C or higher than 30 °C may cause coral bleaching. It will also lead *Octocorallia* corals, which contain symbiotic algae, to bleaching. Soft corals tolerate high and low temperatures better than most Scleractinian coral. However, the optimum water temperature of *Astrogorgia* requires more study in the future.

In this study, the water quality of Qingyu Island and Wuyu Island shows that the sampling depth is 5–7.5 m, the transparency is 0.5–1 m, the temperature is 21–21.7 °C, the pH is 8.2–8.4, the ammonia nitrogen content is less than 0.2 mg/L, and the content of phosphate is 0.05–0.15 mg/L. The water quality and spatial distribution basically reflect the sea environment and land source input. Since the symbiotic algae in *Octocorallia* corals need photosynthesis, these corals are often distributed at a water depth of less than 30 m, such as *Heliopore* and *Tubiporasage* corals. Other corals which do not contain symbiotic algae are often distributed in deeper waters, such as *Dendronephthya* spp., *Scleronephthyasoft* spp., and most *Astrogorgia* spp. [64].

*A. lafoa* and *A. arborea* were first discovered during coral surveys in New Caledonia and the Indian Ocean, respectively. These two species have not been reported before in Xiamen Bay. Whether these two species are native or exotic species needs to be further studied. *A. sinensis* has been previously reported in the waters near Hong Kong and Taiwan, China, but there is no record of this species in Xiamen Bay.

In the future of climate change, the study of the physiology and ecology of the coral reefs will become an important research topic. We should call on human beings to protect the marine environment and create a good habitat for marine animals.

### 4.4. Management Recommendations

Species identification and confirmation are essential basic tasks for diversity management, many of which are not described by science. In this study, newly recorded species enrich the geographic distribution information of *Astrogorgia* and coral species diversity in China. This information will still have to be updated and created in the future. Apparently, management techniques such as stock assessments are characterized by limited data on population dynamics, stock status, and collection efforts. The loss of biodiversity comes from many and complicated reasons, such as environmental wastewater and pollution, illegal fishing, intensive tourism, urbanization, sedimentation, a lack of management measures, and improper management, and primarily from the direct and indirect impacts of climate change, etc. [65]. Strategies have been proposed, including marine-protected areas, rotating closures, prohibited species lists, travel permits, and restricted access to fisheries [66].

To protect coral resources, 11 nature reserves with coral as the main protection object have been established in China [67]. One of them is located in the Dongshan Coral Nature Reserve in Fujian [68]. The establishment of the reserve has effectively protected the precious reef-building stone corals on the northernmost edge of the coast of China. Therefore, it is suggested that coral reserves should be established in the coral distribution and concentration areas in Xiamen Bay. When a coral reserve is established, it can form a network of protected areas with the existing National Endangered Species Reserve and Xiamen National Marine Park. In addition, some measures have been proposed, including strengthening the guidance and management of fishing operations by fishermen in the Gulf, reducing the entry of land-based pollutants, tightening fishery law enforcement and management, and eliminating illegal coral fishing. These implementations will promote the protection of coral species diversity in Xiamen Bay.

## 5. Conclusions

The morphological characteristics of *A. lafoa*, *A. arborea*, and *A. dumbea* have been described in detail, which can provide a reference for further research on the classification and active substances of this genus. There are three species of *Astrogorgia* corals living in Xiamen Bay, namely *A. lafoa*, *A. arborea*, and *A. dumbea*, which are all newly recorded species in Xiamen Bay, of which *A. lafoa* and *A. arborea* are still newly recorded species in the coastal waters of China. The results of this study enrich the geographic distribution information and coral species diversity records of *Astrogorgia* in China and across the world.

**Author Contributions:** Conceptualization, J.-Y.L., T.-J.C., and Y.-P.W.; methodology, J.-Y.L. and Y.-P.W.; software, Y.-P.W., J.-Y.L., J.Y., T.-J.C., and Y.-J.S.; validation, J.-Y.L., Y.-P.W., T.-J.C., and Y.-J.S.; investigation, J.-Y.L., Y.-P.W., J.Y., T.-J.C., and Y.-J.S.; resources, J.-Y.L., Y.-P.W., and J.Y.; data curation, J.-Y.L. and Y.-P.W.; writing—original draft preparation, J.-Y.L., Y.-P.W., and T.-J.C.; writing—review and editing, J.-Y.L. and T.-J.C.; visualization, J.-Y.L., T.-J.C., and Y.-P.W.; supervision, J.-Y.L.; project administration, J.-Y.L.; funding acquisition, J.-Y.L., T.-J.C., and Y.-J.S. All authors have read and agreed to the published version of the manuscript.

**Funding:** This work was supported by the Natural Science Foundation of Fujian Province (No. 2019J01690), the Xiamen Ocean and Fishery Bureau (Southern Center Project) (No. 13GQT001NF14), and the National Foundation Incubation Program of Jimei University (No. ZP2020021). The funders had no role in the study design, data collection, and analysis, the decision to publish, or the preparation of the manuscript.

**Institutional Review Board Statement:** Not applicable.

**Informed Consent Statement:** Not applicable.

**Data Availability Statement:** Not applicable.

**Acknowledgments:** We thank Chang-Feng Dai and Ze-Geng Wu for their suggested comments for the manuscript.

**Conflicts of Interest:** The authors declare that there is no conflict of interest.

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
