# Peer review of "Revealing the Coral Species Diversity in Xiamen Bay: Spatial Distribution of Genus Astrogorgia (Cnidaria, Alcyonacea, Plexauridae) and Newly Recorded Species"

_water, doi:10.3390/w14152417_

Round 1
Reviewer 1 Report
Review of Revealing the coral species diversity in Xiamen Bay
This paper describes the results of marine surveys collecting a 1185 samples across 7 years to specifically focus on the taxonomic distribution of Astrogorgia genera. The paper is generally well written and provides a foundation for future taxonomic work in this area plus a substantial boost for marine conservation planning. I am not an expert in the revision of the Octocorals from genetic information but the pathways described seem to be reasonable and justified.
My comments are relatively minor:
1. Abstract. There are a number of problems here. The statement “…has always been the focus…” is not true. Line 17 has a conflict in stating that the research on Astrogorgia is based on unidentified species but its more complicated than that and based on morphological descriptions etc rather than ‘unidentified’ species. Line 20 – remove ‘were’. To make the abstract independent of the main paper please spell out the acronyms of the places such as QY.
2. Line 106. This sentence needs English correction in several areas.
3. Line 118: “There 17 samples…” does not make sense. Do you mean from the 105 samples collected from the previous sentence?
4. Line 280: change ‘shared’ to share.
5. Line 283 alter the English here to read “…results might be limited by a lack of reference sequences…”
6. Line 300 change ‘are new’ to ‘that are new’
Author Response
We are much grateful for your careful reading of our manuscript and your valuable comments and suggestions to help improve the paper. We have now carefully revised the paper in light of all the comments and suggestions. The following is a point-by-point response.
1. This paper describes the results of marine surveys collecting a 1185 samples across 7 years to specifically focus on the taxonomic distribution of Astrogorgia genera. The paper is generally well written and provides a foundation for future taxonomic work in this area plus a substantial boost for marine conservation planning. I am not an expert in the revision of the Octocorals from genetic information but the pathways described seem to be reasonable and justified.
Answer: We really appreciate some very good and encouraging comments from you.
2. Abstract. There are a number of problems here. The statement “…has always been the focus…” is not true. Line 17 has a conflict in stating that the research on Astrogorgia is based on unidentified species but its more complicated than that and based on morphological descriptions etc rather than ‘unidentified’ species. Line 20 – remove ‘were’. To make the abstract independent of the main paper please spell out the acronyms of the places such as QY.
Answer: We are much grateful for your careful reading of our manuscript and your valuable comments and suggestions to help improve the paper. We have followed your comments, and some adjustments have been made. We fixed the “The focus on coral species diversity cannot be overemphasized”. Line 13-14, and “Most of the current research on the active substances of the genus Astrogorgia is based on unidentified species” Line 16-17, “There are three species of Astrogorgia identificated, including Astrogorgia lafoa, A. arborea and A. dumbea. Among them, A. lafoa, A. arborea are new record species in the waters of China. A. lafoa is distributed in Qingyu Island, A. arborea is distributed in Wuyu Island, A. dumbea is widely distributed in Baiha Reef, Qingyu Island, Wuyu Island and Xiaobai Island”. Line 20-22.
3. Line 106. This sentence needs English correction in several areas.
Answer: we have followed your comments, and some adjustments have been made. We fixed the “This study mainly used external morphology and gene fragment technology to classify and identify coral samples collected from the waters in Xiamen Bay. The aim of this research is to explore the geographic distribution and coral species diversity of Astrogorgia. The result can provide references for in-depth on the classification and suggestions for formulating management policies and the conservation of coral resources”. Line 106-109.
4. Line 118: “There 17 samples…” does not make sense. Do you mean from the 105 samples collected from the previous sentence?
Answer: Thank you very much for your valuable advice, which means that out of 1185 samples, 17 samples belong to the genus Astrogorgia.
5. Line 285: change ‘shared’ to share.
Answer: We have followed your comments, and have corrected this error.
6. Line 288 alter the English here to read “…results might be limited by a lack of reference sequences…”
Answer: we have followed your comments, and some adjustments have been made. We fixed the “the results may be limited by lack reference sequences of congener species in this study”. Line 288.
7. Line 305 change ‘are new’ to ‘that are new’
Answer: We have followed your comments, and have corrected this error. We fixed the “There are three species that are new record species of Astrogorgia in Xiamen Bay”, Line 305.

Reviewer 2 Report
After read and review the manuscript, the authors using external morphology and gene fragment technology to classify and identify coral samples collected from the waters in Xiamen Bay, China, to explore the geographic distribution and coral species diversity of genus Astrogorgia, in order to provide references for an in-depth on the classification and suggestions for formulating management policies and the conservation of coral resources.
I found the manuscript interesting, and with an important contribution to the knowledge of the geographical distribution of the Astrogorgia genus in Xiamen Bay and in China. I have few comments which I describe in detail below.
Line 57: Change Seleactinia by Scleractinia, and Hexacorollia by Hexacorallia
Line 62: Change Astrogrogia by Astrogorgia
In the aim of your research, line 106 to 109, you wrote: “The aim of this research, which explore the geographic distribution and coral species diversity of Astrogorgia, is to provide references for in-depth on the classification and suggestions for formulating management policies and the conservation of coral resources.” However, I did not read any suggestion for formulating management policies and conservation of coral resources along the manuscript, and as a part of the aim of your work you need to give some arguments and suggestion to finish it.
Section 2. Materials and Methods
Subsection 2.1. Sample collection and study area.
You only detail the sampling sites and times but do not mention anything about how you preserved and transported the samples to the laboratory, specifically for the molecular analysis. Please explain.
Line 300: You wrote “There are three species are new record species of…” correct please
Line 302: Change neretic by neritic
Line 398: What is an octopus coral?
Author Response
We are much grateful for your careful reading of our manuscript and your valuable comments and suggestions to help improve the paper. We have now carefully revised the paper in light of all the comments and suggestions. The following is a point-by-point response.
1. I found the manuscript interesting, and with an important contribution to the knowledge of the geographical distribution of the Astrogorgia genus in Xiamen Bay and in China. I have few comments which I describe in detail below. Line 57: Change Seleactinia by Scleractinia, and Hexacorollia by Hexacorallia
Answer: We really appreciate some very good and encouraging comments from you. We have followed your comments, and have corrected this error.
2. Line 62: Change Astrogrogia by Astrogorgia
Answer: We have followed your comments, and have corrected this error.
3. In the aim of your research, line 106 to 109, you wrote: “The aim of this research, which explore the geographic distribution and coral species diversity of Astrogorgia, is to provide references for in-depth on the classification and suggestions for formulating management policies and the conservation of coral resources.” However, I did not read any suggestion for formulating management policies and conservation of coral resources along the manuscript, and as a part of the aim of your work you need to give some arguments and suggestion to finish it.
Answer: we have followed your comments, and some adjustments have been made.
We fixed the “4.4. Management recommendations
Species identification and confirmation are essential basic tasks for diversity management, many of which are not described by science. In this study, newly rec-orded species enrich the geographic distribution information of Astrogorgia and coral species diversity in China. This information will still have to be updated and created in the future. Apparently, management techniques such as stock assessments which are characterized by limited data on population dynamics, stock status, and collection ef-fort. The loss of biodiversity comes from many and complicated reasons, such as envi-ronmental waste water and pollution, illegal fishing, intensive tourism, urbanization, sedimentation, lack of management measures and improper management, and primar-ily from the direct and indirect impacts of climate change. etc [65]. Strategies have been proposed including marine protected areas, rotating closures, prohibited species lists, travel permits and restricted access to fisheries [66].
In order to protect coral resources, 11 nature reserves with coral as the main pro-tection object have been established in China [67]. One of them is located in the Dongshan Coral Nature Reserve in Fujian [68]. The establishment of the reserve has effectively protected the precious reef building stone coral on the northernmost edge of the coast of China. Therefore, it is suggested that coral reserves should be established in the coral distribution and concentration areas in Xiamen Bay. When a coral reserve is established, it can form a network of protected areas with the existing National Endangered Species Reserve and Xiamen National Marine Park. In addition, some measures have been pro-posed, including strengthening the guidance and management of fishing operations by fishermen in the Gulf, reducing the entry of land-based pollutants, tightening fishery law enforcement and management, and eliminating illegal coral fishing. These implementations will promote the protection of coral species diversity in Xiamen Bay”. Line 424-448.
4. Section 2. Materials and Methods
Subsection 2.1. Sample collection and study area.
You only detail the sampling sites and times but do not mention anything about how you preserved and transported the samples to the laboratory, specifically for the molecular analysis. Please explain.
Answer: We have followed your comments, and some adjustments have been made. We fixed the “Live coral samples were collected and brought back to the laboratory, where they were numbered and photographed. The samples were stored in 95% ethanol solution or -20℃ refrigerator. Subsequently, these samples were used for microscopic morphology observation and molecular biological analysis. The processed samples were deposited in the Coral Ecology Laboratory of Fisheries College, Jimei University.” Line 123-127.
5. Line 305: You wrote “There are three species are new record species of…” correct please
Answer: We have followed your comments, and have corrected this error. We fixed the “There are three species that are new record species of Astrogorgia in Xiamen Bay, including A. lafoa, A. arborea and A. dumbea. ” Line 305-306.
6. Line 307: Change neretic by neritic
Answer: We have followed your comments, and have corrected this error.
7. Line 403: What is an octopus coral?
Answer: We have followed your comments, and some adjustments have been made. We fixed the “It will also lead Octocorallia corals which contained symbiotic algae to bleaching. ” Line 402-403.

Round 2
Reviewer 2 Report
After reviewing the manuscript, I confirm that the suggested corrections have been satisfactorily made, for which I have no problem with the manuscript being accepted for publication. I only have two minimum observations that I list below:
Line 433: you wrote "...climate change. etc [65]. " Please change dot after change by a comma.
Line 439: You wrote "stone coral", please change by stony corals